# Assessing Cognitive Workload Using Cardiovascular Measures and Voice

**DOI:** 10.3390/s22186894

**Published:** 2022-09-13

**Authors:** Eydis H. Magnusdottir, Kamilla R. Johannsdottir, Arnab Majumdar, Jon Gudnason

**Affiliations:** 1Center for Analysis and Design of Intelligent Agents, Reykjavik University, 101 Reykjavik, Iceland; 2The Lloyd’s Register Foundation Transport Risk Management Centre, Imperial College, London SW7 2AZ, UK

**Keywords:** cognitive workload, cardiovascular signals, speech processing, emotion recognition

## Abstract

Monitoring cognitive workload has the potential to improve both the performance and fidelity of human decision making. However, previous efforts towards discriminating further than binary levels (e.g., low/high or neutral/high) in cognitive workload classification have not been successful. This lack of sensitivity in cognitive workload measurements might be due to individual differences as well as inadequate methodology used to analyse the measured signal. In this paper, a method that combines the speech signal with cardiovascular measurements for screen and heartbeat classification is introduced. For validation, speech and cardiovascular signals from 97 university participants and 20 airline pilot participants were collected while cognitive stimuli of varying difficulty level were induced with the Stroop colour/word test. For the trinary classification scheme (low, medium, high cognitive workload) the prominent result using classifiers trained on each participant achieved 15.17 ± 0.79% and 17.38 ± 1.85% average misclassification rates indicating good discrimination at three levels of cognitive workload. Combining cardiovascular and speech measures synchronized to each heartbeat and consolidated with short-term dynamic measures might therefore provide enhanced sensitivity in cognitive workload monitoring. The results show that the influence of individual differences is a limiting factor for a generic classification and highlights the need for research to focus on methods that incorporate individual differences to achieve even better results. This method can potentially be used to measure and monitor workload in real time in operational environments.

## 1. Introduction

The cognitive workload of personnel in a workforce, especially those involved in safety critical industries, e.g., airline pilots and air-traffic controllers, is crucial to ensuring both the well being and productivity of the personnel and the broader safety of the public. Consideration of workload and its management is therefore a crucial aspect of any safety management system in a safety critical organisation. Assessing and monitoring cognitive workload is, therefore, of great importance. While there are numerous methods by which to measure workload, e.g., subjective methods, psychophysiological approaches to cognitive workload monitoring, that use signals such as cardiovascular measures and electroencephalography (EEG), have recently shown promise in identifying cognitive workload in laboratory settings [1,2,3,4,5,6]. Increased mental demand is highly correlated with increased cardiovascular reactivity [7,8]. Various classifier methods have also been used to recognize different cognitive workload states (reliably high and low workload) based on combined psychophysiological signal sources [9,10,11]. Despite this, going beyond a binary high/low workload classification has proved to be difficult. The problem might at least partly be methodological. Many current approaches to monitoring cognitive workload fail to consider the individual variation in responses to workload and this shortcoming has been highlighted in the literature [12], though attempts have been made by the authors to address this issue specifically for cardiovascular measures in [13,14]. Furthermore, prior work has not taken into consideration combining the cardiovascular signal with another promising signal source, the individual’s speech even though studies suggest that the speech signal may be a good indication of the individual’s mental state (e.g., see [15,16]).

This paper shows how cognitive workload can be assessed and monitored using cardiovascular measures and voice. Heart rate and blood pressure signals are combined with voice signal formant features to classify cognitive workload into three levels of difficulty. The feature level fusion combination is carried out using each heartbeat to synchronize the two data sources. This allows the subsequent machine learning mechanism to monitor cognitive workload on a beat-by-beat basis. This method is evaluated using two cohorts of 97 volunteers and 20 airline pilots, respectively, each of which conducted out over 45 min of cognitive tasks while being recorded and monitored.

The rest of the paper is organised as follows. The premise of the paper is laid out in Section 1 where speech and cardiovascular measures are described with respect to cognitive workload along with relevant work in the area. The main challenges of assessing cognitive workload using these measures is described and our approach is introduced. In Section 2 the experimental methodology is described and Section 3 explains the data processing. The results of the experiments are laid out in Section 4. The results are summarised and discussed in Section 5, connected to prior work in the area and their main strength and weaknesses highlighted. Overall conclusions are drawn in Section 6.

### 1.1. Cardiovascular Measures and Speech

Cardiovascular measures are relatively unobtrusive and well suited for the aviation environment where speech communications is used to solve tasks. Furthermore, with technological enhancements such as wearable devices, these measures are now becoming even less intrusive than before. Short term fluctuations in task demand affects the cardiovascular system [7,8] and these responses can be objectively identified by monitoring the cardiac muscle or the vascular system by observing, for example, the heart rate (HR) and the blood pressure and stroke volume [17,18].

An alternative way of monitoring cognitive workload is through the speech signal. Whilst not applicable in all environments, it may be ideal in situations where speech can be captured and communications can be monitored in real-time without interruptions. Yin et al. [15] performed a trinary classification task using Mel-frequency cepstrum coefficients and prosodic features with a speaker adapted Gaussian mixture model. This feature extraction method was extended [19] to include targeted extraction of vocal tract features through spectral centroid frequency and amplitude features. In both instances, a relatively small data set was used for validation where each participant performed reading and Stroop tasks. These feature extraction and classification schemes indicated a strong relationship between cognitive workload and the speech signal within the experimental framework of the studies.

Cognitive workload experiments using speech have been carried out for real-life tasks in military flight simulation [20]. This approach has the advantage of being closer to real operational situations indicating that the technology is suitable for aviation applications. Mean change in fundamental frequency and speech intensity was used to detect cognitive workload of the participants, but more detailed speech analysis was not performed. Speech analysis has also been used for related tasks of affective speech classification. For example, the Interspeech 2014 Cognitive load challenge (Computational Paralinguistics ChallengE, ComParE) [21] was based on the same principle. A data set of 26 participants provided speech recordings and EEG signals during a reading task and a low-, medium-, and high cognitive load level Stroop tasks. The winning entry used an i-vector classification scheme based on a combined feature set of fused speech streams, prosody and phone-rate [22].

Combining different physiological signals may provide a more prominent, detailed cognitive workload monitoring tool [23]. Most commonly, studies focus on cardiovascular signals combined with electrical brain activity signals either as a pair [24,25] or grouped with signals such as galvanic skin response [26] or oculomotor measures [9,27]. No attempts, to our knowledge, have been reported investigating the supplemental possibility of cardiovascular and speech signals for cognitive workload monitoring.

### 1.2. Related Work on Cognitive Workload Classification

Various pattern recognition methods have been applied to the task of classifying cognitive workload using psychophysiological measures. It has been pointed out that artificial neural networks are opaque and hard to interpret in terms of how individual variables interact to predict workload [11]. Classification methods have been used such as discriminant analysis and support vector machines [28], as well as logistic regression and classification trees [11]. There is no indication that other classification methods can provide better results for cognitive workload monitoring [11,28]. Recently, however, trinary cognitive workload level classification with cardiovascular signals has been demonstrated with promising results [29].

Few studies have used artificial neural networks to classify cognitive workload states in the field of air-traffic control using combined physiological signals [11,30,31,32,33]. In particular, multiple psychophysiological measures were combined to provide high accuracy in classifying at least a limited number of cognitive load states [31,33]. High binary classification accuracy was achieved for high and low workload states in air-traffic control using neural networks based on EEG and electrocardiography (ECG) signals [33]. However, when the training scenario included four and seven different cognitive load states based both the on complexity and the number of aircraft, the classifier confused adjacent states and was unable to distinguishing between low and medium or medium and high states. A neural network model based on multiple EEG channels, HR, and eye-blink measures produced reliable discrimination between low and high workload and was also able to distinguish between two out of three load-tasks [32] and neural networks also performed well in distinguishing high and low workload particularly at small time intervals [11]. It was pointed out however, that whilst promising, EEG is both complex to use and not easily portable [34]. This work presented a neural network trained on various cardiovascular measures along with performance-based measures and did not manage to reliably classify different cognitive workload states. Other examples of multi-modal fusion for cognitive workload assessments can be found in [35,36].

### 1.3. Challenges in Assessing Cognitive Workload

The main challenge of assessing cognitive workload is the latent nature of the variable in question. A close proxy of cognitive workload is the task difficulty which is typically used when assessing cognitive workload. Albeit close, the relationship between task difficulty and cognitive workload is complex depending on issues spanning from the nature of the task to the condition of the individual being assessed. Tasks can rely on one or more senses (e.g., sight and hearing) and require one or more motor skills (e.g., touch and voice) and be simple or complex in space and/or time. The condition of the individual brings other variables such as ability and fatigue into the equation.

It has also been known for quite some time now that individuals show different psychophysiological responses to cognitive workload. Measured voice parameters, for example, were found to be different between individuals with respect to workload as far back as 1968 [37]. The matter of individual differences has been noted periodically with Ruiz et al. [38], for example, claiming that more than a single voice parameter needed to be measured as an indication of workload and Grassmann et al. [12] found that integrating individual differences may reduce unexpected variance in workload assessment. Moreover, research has shown that individual working memory capacity may play a critical role in determining how individuals react to changes in cognitive workload [14].

Cognitive workload is also perceived to be a continuous variable although its effect on the individual might be categorical (i.e., fight-or-flight vs. rest-and-digest). Researchers have, however, struggled with this assessment and many have reduced the problem of cognitive workload monitoring to a binary classification of high or low workload [31,33]. How cognitive workload assessment is developed beyond this dichotomy remains an open research question.

Measuring and combining many different psychophysiological measures also presents a set of challenges that researchers have grappled with [39]. Cognitive workload presents differently through the systems being measured (e.g., heart-rate, speech or the brain’s electrical activity) so making more than one of these data sources available for the assessment should make the monitoring more robust and accurate. The most straightforward method of combining feature sets from different sources would be simply to concatenate them. There are, however, a few issues that need to be addressed before a concatenated feature set can be successfully used as a pattern recognition classification input. The sampling rates of the two or more signals might not be the same hence some sort of resampling and alignment needs to take place. Alignment in time has to be ensured during data recordings as well as their correspondence after individual feature extraction is concluded.

### 1.4. Our Approach

Most of these highlighted challenges are addressed in our approach. We use task difficulty as a proxy to cognitive workload but we make an effort in keeping the tasks sufficiently simple so that their difficulty can be interpreted as cognitive workload. Furthermore, we train a cognitive workload classifier for each participant separately to meet the challenge of individual differences in responses to cognitive workload. The classifiers are based on three classes: low, medium and high workload in an attempt to move beyond binary classification. We also present a novel feature level fusion approach between non-intrusive cardiovascular measures and voice, thereby improving the accuracy of the cognitive workload monitoring.

## 2. Cognitive Workload Experiments

The cognitive workload experiments are set up so that the relationship between task difficulty and workload is close and with three difficulty levels to reflect the non-binary approach taken. The amount of data obtained from each individual should be sufficient in order to model each participant’s response to cognitive workload separately. The data collected are cardiovascular data and voice recordings, analysed separately and combined in a trinary classification of cognitive workload.

### 2.1. Experiment Design and Configuration of Tasks

Figure 1 depicts a chart of the progress of tasks, instructions, self-assessment questionnaires and resting periods implemented in the experiment. The flow chart depicts all elements included in the experiment such as the OSPAN test (see [40]) and reading task, but the focus in this particular paper is on the Stroop tasks.

Cognitive workload levels was introduced through the well-established cognitive stimuli word/color task published by Stroop in 1935 [41]. Throughout the Stroop task a set of either incongruent (e.g., red in blue color) or congruent (e.g., red in red color) color names appears in front of the participant. In this specific setup the Icelandic color names Blue, Green, Brown, Red and Pink were used with the last color name of each set always being Black, with 36 (35 + 1) words appearing in a 6 × 6 matrix on each screen. This design was included to indicate to the researcher, controlling the flow of the screens, that the participant had finished the current screen *j*. The participant’s assignment was to say the colors of the words aloud but not to read the words (of the colors). Three cognitive workload levels were induced with the settings of congruence, incongruence and time limits as follows:Level 1—Seven congruent sets of screens with all 36 color names appearing on the computer screen at the same time.Level 2—Six sets of screens with alternating incongruence levels of 0.3 and 0.7 with all 36 color names appearing on the computer screen at the same time.Level 3—Eight sets of screens with one word appearing at a time at randomly timed intervals of 0.75 s and 0.65 s. Here, the same incongruence set-up was applied as in Level 2 and the same number (36) of color names as in Level 1 and 2.

The number of screens in each cognitive workload level were chosen in advance to ensure approximately the same duration of levels. The cognitive workload levels were introduced in six different orders to the participants using the Latin square technique.

The participants were introduced to the task by having them read detailed instructions, appearing on the computer screen, aloud. As depicted in Figure 1, each of the three Stroop sessions contained the cognitive workload task, self-assessment questionnaire and resting period, repeated three times for each level with the total number of $J_{p}=21$ screens, for each participant *p* and the screen index $j=\{1,2,\cdots ,J_{p}\}$.

The different resting periods and their strategic positions are shown in Figure 1. These periods were designed to ensure that the participant had sufficient time to recover between tasks and reduce its influence on succeeding tasks.

Participants in the experiment performed on the operation span task (OSPAN). The OSPAN task is a working memory task that measures the working memory span by having participants solve simple equations and remembering a word at the same time [42]. In this task, participants read out loud an equation (e.g., is (8×3)+2=25) and answer whether the equation is correct or incorrect. The equation is followed by a word (e.g., car) that also is read out loud. There are 12 sets of 3 × 2 words/eq, 3 × 3 words/eq, 3 × 4 words/eq and 3 × 5 words/eq in total. The participants’ task is to remember the presented words in the correct order for each set. The total number of words to be remembered is 42; hence, the participants could get a maximum score of 42 and a minimum score of 0. One point was given for a correct word in the correct order and higher scores indicate higher working memory capacity. The results for the OSPAN task were not used in the current paper.

### 2.2. Two Cohorts

The method was developed on two sets of participants: volunteers who visited the laboratory of Reykjavik University (university cohort) and pilots from a commercial airline, Icelandair, that had just completed a simulation exercise at TRU Flight Training Iceland (pilot cohort). The university cohort had a total number of $P_{1}=97$ participants with average age of 25.2 ± 5.78 and a gender ratio of 27.83% male to 72.17% females. The pilot cohort had a total number of $P_{2}=20$ participants with average age of 41.35 ± 9.36 and a gender ratio of 90% male to 10% females.

## 3. Data Processing

### 3.1. Cardiovascular Measures

The equipment used to record cardiovascular signals was the Finometer Pro from Finapres, capturing a complete hemodynamic profile of the participants cardiovascular responses [43,44]. There has been extensive validation of Finometer measurements, including studies involving young participants (e.g., [45,46]). With the Finapres the signals are obtained using a finger cuff and an upper arm cuff for calibration of the reconstructed blood pressure. Although cardiovascular signals are sampled continuously by the Finometer, calculations for a number of relevant measures are performed for each heartbeat *n*, making the output measures consistently uneven in time. The following ten measures for each *n* were obtained from the output of the Finometer Pro system: (1) heart rate, (2) systolic pressure, (3) diastolic pressure, (4) mean pressure, (5) pulse interval, (6) stroke volume, (7) left ventricular pressure energy, (8) cardiac output, (9) total peripheral resistance, and (10) maximum steepness and are designated as $c_{n,i}$ where i∈{1,2,⋯,10} is the index for the measure and *n* is the integer time index of the heartbeat. The ten dimensional feature vector for the *n*-th heartbeat is therefore
(1)$$c_{n}={\left[c_{n,1},c_{n,2},\cdots ,c_{n,10}\right]}^{T}$$
and an $N_{j}\times 10$ data matrix $C_{j}={\left[c_{1},c_{2},\cdots ,c_{N_{j}}\right]}^{T}$ where $N_{j}$ is the total number of heartbeats in the segment. The screen index *j* is sometimes dropped whenever it is clear that we are talking about a data matrix for a specific screen, so $C_{j}=C$.

### 3.2. Speech Signal Processing

All speech was recorded during the experiment by using both a head-mounted and a table-top microphone. The recordings were stored in linear pulse code modulation files with 48 kHz sampling frequency and 16 bits per sample. Each file corresponds to the present screen of the experiment and the recordings during the J=21 Stroop screens were processed further for classification.

There are numerous means of extracting useful features from speech recordings. The aim in this work is to characterize the voice without especially targeting linguistic or prosodic content of the speech. One way of doing that is to rely on traditional acoustic features such as those based on the short time frequency transform of the signal. In this paper, however, we track the first three formant with respect to time. The formants are directly related to the articulation of the vocal tract and can therefore be interpreted directly. Formant tracks have also been used successfully as features before, for example when modelling emotion and depression [47]. This method was adopted in our previous work for cognitive workload monitoring in [16].

The formant features were extracted using the Kalman-based auto-regressive moving average smoothing (KARMA) algorithm [48] as in [47]. The main advantage of using KARMA is that the algorithm produces smoother formant tracks than other methods and it provides a sensible interpolation during non-voiced periods. The algorithm was configured to extract three formants, three anti-formants and bandwidths every 10 ms. Only the three formant frequencies $f_{1,l},f_{2,l},f_{3,l}$ were used to produce a feature vector
(2)$$f_{l}={\left[f_{1,l},f_{2,l},f_{3,l}\right]}^{T}$$
for the *l*-th frame in the speech segment (screen) and an $L_{j}\times 3$ data matrix $F=F_{j}={\left[f_{1},f_{2},...,f_{L_{j}}\right]}^{T}$ where $L_{j}$ is the total number of frames in the segment.

### 3.3. Feature Extraction, Synchronization and Fusion

The cardiovascular features $c_{n}$ and the formant features $f_{l}$ are processed further to enrich the representation with a dynamical context and to fuse these two information sources together for a joint classification. Figure 2 depicts the feature extraction, synchronization process, feature set combinations and classification schemes introduced. A richer dynamical context is achieved by calculating the so-called delta and acceleration features and the feature level fusion is attained by calculating formant statistics around each heartbeat. This is outlined below.

#### 3.3.1. Delta and Acceleration Features

Delta and acceleration features were calculated on the static cardiovascular vector $c_{n}$ and the formant feature vector $f_{l}$ to include information about the rate of change. For an arbitrary feature vector $v_{r}$ whose feature index is denoted with *i* and time index with *r* the coefficients of the delta vector $\delta_{r}$ are calculated from the measure $v_{r,i}$ along the time index *i* by using
(3)$$\delta_{r,i}=\frac{\sum_{m=1}^{M}m\left(v_{r+m,i}-v_{r-m,i}\right)}{2\sum_{m=1}^{M}m^{2}},$$
where *M* denotes the number of adjacent features before and after *r* used to derive the delta features. This is set to M=2 for this paper (resulting in a window size of 5). The delta vector is therefore obtained by $\delta_{r}={\left[\delta_{r,1},\delta_{r,2},\cdots \right]}^{T}$ and the acceleration feature vector $a_{r}$ is calculated using the same formula but using $\delta_{r,i}$ instead of $v_{r,i}$. The delta and acceleration vectors are then appended to the feature vector to produce $v_{r}^{\left(\delta ,a\right)}={\left[v_{r}^{T},\delta_{r}^{T},a_{r}^{T}\right]}^{T}$. Hence, a delta and acceleration extended cardiovascular feature vector would be denoted $c_{n}^{\left(\delta ,a\right)}={\left[c_{n}^{T},\delta_{n}^{T},a_{n}^{T}\right]}^{T}$ and its corresponding data matrix $C^{\left(\delta ,a\right)}$; for the formant feature vector the extension is $f_{l}^{\left(\delta ,a\right)}={\left[f_{l}^{T},\delta_{l}^{T},a_{l}^{T}\right]}^{T}$ and its corresponding data matrix $F^{\left(\delta ,a\right)}$.

#### 3.3.2. Synchronizing Formant Features and Heartbeats

Each audio recording is of a single screen *j* containing $N_{j}$ heartbeats and $L_{j}$ formant frames. The heartbeats occur at $\tau_{n}$ seconds into the recordings and each pulse interval is therefore $\Delta_{n}=\tau_{n}-\tau_{n-1}$. The start of each formant frame occurs at $t_{l}$ and the fixed formant frame increment is $t_{l}-t_{l-1}=10$ ms. A typical pulse interval is $\Delta_{n}=700ms$ (c.a. 86 beats-per-minute) which gives a ratio of 70:1 formant frames to heartbeat. Figure 3 depicts the synchronization process graphically with heartbeats (stem) and the vertical dotted lines mark the number of formant frames attached to it.

The formant feature data matrix F is divided into $N_{j}$ sub-matrices $F_{n}$ which correspond to each heartbeat. The matrix $F_{n}={\left[f_{1},f_{2},\cdots ,f_{M_{n}}\right]}^{T}$ has the row vector $f_{m}$ where $m\in \{l|t_{l}>\tau_{n}-\frac{1}{2}\Delta_{n},t_{l}<\tau_{n}+\frac{1}{2}\Delta_{n+1}\}$. The three columns in $F_{n}$ correspond to three formant tracks in the neighbourhood of the *n*-th heartbeat. Ten features are calculated for each of the three formant tracks so for the first formant track, the first three are the coefficients of a fitted second order polynomial, denoted as $\phi_{1,n}$, $\phi_{2,n}$, and $\phi_{3,n}$ and then the minimum $\phi_{4,n}$ and maximum $\phi_{5,n}$ values, mean $\phi_{6,n}$, median $\phi_{7,n}$, standard deviation $\phi_{8,n}$, skewness $\phi_{9,n}$ and kurtosis $\phi_{10,n}$. The same features are calculated for the second $\phi_{11-20,n}$ and third formant tracks $\phi_{21-30,n}$. The corresponding feature vector is
(4)$$\phi_{n}={\left[\phi_{1,n},\phi_{2,n},\cdots ,\phi_{30,n}\right]}^{T}$$
and the $N_{j}\times 30$ dimensional data matrix for the segment is $\Phi ={\left[\phi_{1},\phi_{2},...,\phi_{N_{j}}\right]}^{T}$.

Similarly, the delta and acceleration expanded formant data matrix $F^{\left(\delta ,a\right)}$ is divided up into sub-matrices $F_{n}^{\left(\delta ,a\right)}$ in the same way to produce three formant tracks, three delta formant tracks and three acceleration formant tracks in the neighbourhood of the *n*-th heartbeat. The same features are calculated for each of the nine tracks to produce a ninety-dimensional feature vector
(5)$$\gamma_{n}={\left[\gamma_{1,n},\gamma_{2,n},\cdots ,\gamma_{90,n}\right]}^{T}$$
and an $N_{j}\times 90$ dimensional data matrix for the segment $\Gamma ={\left[\gamma_{1},\gamma_{2},...,\gamma_{N_{j}}\right]}^{T}$. Notice that the delta and acceleration operators are applied before the heartbeat synchronized features are calculated and could be reapplied again to produce the data matrices $\Phi^{\left(\delta ,a\right)}$ and $\Gamma^{\left(\delta ,a\right)}$ but this was not conducted in this work.

#### 3.3.3. Data Fusion

The four basic feature vectors are the cardiovascular vectors without ($c_{n}$) or with ($c_{n}^{\left(\delta ,a\right)}$) delta and acceleration features; and synchronised formant features without ($\phi_{n}$) or with ($\gamma_{n}$) delta and acceleration features can now be concatenated as they have been synchronised in time (the *n* refers to the same time instant or heart-beat). We concatenate these features in four different ways to create the following feature vectors for feature level fusion [49]: $x_{n}={\left[\phi_{n}^{T},c_{n}^{T}\right]}^{T}$, $x_{n}={\left[\phi_{n}^{T},c_{n}^{(\delta ,a)T}\right]}^{T}$, $x_{n}={\left[\gamma_{n}^{T},c_{n}^{T}\right]}^{T}$ or $x_{n}={\left[\gamma_{n}^{T},c_{n}^{(\delta ,a)T}\right]}^{T}$. The symbol $x_{n}$ is used as a generic feature vector at heartbeat *n* in the following discussion. In feature-level fusion the subsequent classifier models the concatenated vectors jointly and delivers a classification result based on both vectors simultaneously.

### 3.4. Sequence vs. Beat Classification

The classifiers designed and implemented in this paper are based on the feature sets containing $\phi_{n}$ and $c_{n}$ and their time dynamic counterparts $\phi_{n}^{\left(\delta ,a\right)}$ and $\gamma_{n}$. All of which are evaluated in two types of classifiable entities, either (1) from a single heartbeat feature vector $x_{n}$ or (2) a sequence of heartbeats from a single screen represented with the data matrix $X={\left[x_{1},x_{2},\cdots ,x_{N}\right]}^{T}$.

The classifiers’ soft (likelihood) scores were retrieved and are denoted either as $y_{k}\left(x_{n}\right)$ for the single heartbeat classifier or $y_{k}\left(X\right)$ for the sequence classifier. The index k∈{1,2,3} denotes Stroop level one, two or three, respectively. The vector,
(6)$$y\left(x_{n}\right)={\left[y_{1}\left(x_{n}\right),y_{2}\left(x_{n}\right),y_{3}\left(x_{n}\right)\right]}^{T}$$
contains the soft scores for the heartbeat at time index *n* and the heartbeat classification simply chooses the class with the maximum value in that vector for each *n*.

For the sequence classification, the beat-based soft scores are collected in an N×K output matrix
(7)$$Y={\left[y^{T}\left(x_{1}\right),y^{T}\left(x_{2}\right),\cdots ,y^{T}\left(x_{N}\right)\right]}^{T}$$
and summed over each class column to obtain $y_{k}\left(X\right)$ sequence-based soft score vector (with k∈{1,2,3}),
(8)$$y\left(X\right)={\left[y_{1}\left(X\right),y_{2}\left(X\right),y_{3}\left(X\right)\right]}^{T}$$
whose maximum is chosen as the classified level.

### 3.5. Classifier Architectures

Two very different supervised learning classifier architectures were used to evaluate this method, both implemented using the statistics toolbox in Matlab: support vector machine (SVM), $y_{SVM}\left(x_{n}\right)$ and random forests (RF), $y_{RF}\left(x_{n}\right)$. This was conducted to see to what extent the results depend on the choice of classifier architecture.

A support vector machine is fundamentally a binary classifier and therefore to solve the trinary classification problem, three two-class one-vs.-rest binary SVM classifiers were implemented, one for each Stroop level. The soft score output $y_{k}\left(x_{n}\right)$ is the signed distance from the decision boundary for each of the three classifiers where k∈{1,2,3}. The class is then determined by the one-vs.-rest classifier which obtains the maximum signed distance from the decision boundary. If all scores are negative, then the class closest to the decision boundary with the least negative score is chosen. The training was conducted using the default linear kernel function and the predictors were standardized before training.

The random forest classifier was trained for each heartbeat using one hundred decision trees. The minimum number of observations in a leaf was set to one. The soft score $y_{k}\left(x_{n}\right)$ for the random forest classifier is the proportion of trees in the ensemble predicting class *k* and is interpreted as the probability of this observation $x_{n}$ originating from this class.

### 3.6. Classifier Training and Evaluation

A participant dependent classification was implemented in this work where a leave-one-out training strategy was used to evaluate the performance of the feature sets. For the 21 screens for each participant, one screen was left out while the other 20 screens were used for training. Repeating this process 21 times ensured classification results for all screens based on a model trained with the bulk of the available data.

When using an entire screen as a test token this would lead to a single result that could be compared to the actual class of the screen producing a single classification result.

Using an entire screen as a test token would lead to a single result that can be compared to the actual class of the screen producing a single classification result. When using each heartbeat as a test token the number of results amounted to the number of heartbeats in the screen that was left out (for example for a test screen of 30 s duration with an average heartbeat of 70 beats-per-minute (bpm) we have a total of 35 heartbeats). The leave-one-out procedure (repeated 21 times) was collected in confusion matrices and the results from the leading method is presented in an aggregated form (see tables in Section 4.3). The test set misclassification rate (MCR) as reported as a percentage in the result section is the average MCR over the entire $P_{1}=97$ and $P_{2}=20$ participant data sets, respectively, (see tables in Section 4.1 and Section 4.2). The standard error reported alongside the MCR reflects the confidence in the average MCR estimate and should not be confused with the larger standard deviation of MCR over the population of participants.

## 4. Results

The first two subsections demonstrate how we use the voice and cardiovascular measures to classify cognitive workload into low, medium and high levels based on single heartbeat and a sequence of heartbeats, respectively. We then analyse the results in more detail for the best of classifier/feature set set-up demonstrating how the class specific misclassification rates and mistrust rates are distributed from low to high cognitive workload level. Here, we also demonstrate how the pilot cohort is similar to the university cohort.

### 4.1. Single Heartbeat Classification

The results based on a single heartbeat is shown in Table 1 for the university cohort. The two columns show the misclassification rates for each classifier and each line provides the results for the features used for the classification. The results above the horizontal line are for the two information sources (voice and cardiovascular measures) with or without dynamic features. Below the horizontal line are the results for the combined feature set. For example the $x_{n}=\gamma_{n}$ is the feature set derived from the formant tracks and their dynamic features achieves 57.88±0.18% misclassification rate for the SVM classifier and 52.58±0.18% for the random forest classifier. These particular results are not very good since chance is 66.66% for the trinary low, medium, high cognitive workload classification. The best results obtained for a single information source (highlighted in bold above the line) is achieved by using the SVM classifier and the cardiovascular features with dynamic features $x_{n}=c_{n}^{\left(\delta ,a\right)}$. The best results overall (also highlighted) is obtained by the SVM classifier using a combined feature sets of voice and cardiovascular features and dynamic features, i.e., $x_{n}$ is created by concatenating $\gamma_{n}$ and $c_{n}^{\left(\delta ,a\right)}$.

From these results one can see that the SVM classifier performs better except for when voice is used on its own when the RF classifier is better. In addition, when using voice alone, adding dynamic features provides significant advantage but only a small advantage is obtained when the cardiovascular features are used either alone or fused with the voice features. Out of context, the best results of 32.14±0.17% does not sound good for a trinary classification but when taking into account that the decisions are only based on measurements around a single heart beat, this has to be considered very good. This optimum classifier method using SVM and the $\gamma_{n}$ and $c_{n}^{\left(\delta ,a\right)}$ features was applied to the pilot cohort (of $P_{2}=20$ participants). The misclassification rate for this experiment is 34.49±0.40%.

### 4.2. Heartbeat Sequence Classification

The results based on a sequence of heart beats corresponding to a single Stroop screen is shown in Table 2 for the university cohort. The table is set up in the same way as Table 1 with the classifier results in the two columns each line corresponding to a choice features and the horizontal line divides the results up into single feature set vs combined ones. Now for single information stream, the best result of 22.97±0.93% is achieved by the RF classifier using cardiovascular features without dynamic features C. The best results of 15.17±0.79% for combination of features is achieved using the SVM classifier when fusing the Γ and the $C^{\left(\delta ,a\right)}$ feature sets. This optimum classifier method was also applied to the pilot cohort where the misclassification rate was 17.38±1.85%.

The difference between the SVM and RF classifiers is not as clear for the heartbeat sequence classification. The RF classifier performs better when the single information source is used whereas the SVM classifier performs better for the fused data sets. The addition of dynamic features does improve performance but the MCR reduction is rarely above 1%.

### 4.3. Trinary Classification for University and Pilot Cohorts

Table 3 and Table 4 show the confusion tables for the heart beat sequence classifiers using the SVM classifier and the combined formant tracks Γ and cardiovascular features $C^{\left(\delta ,a\right)}$ for the University and pilot cohorts, respectively. The tables display the classification counts of each experiments as well as the misclassification rate (MCR, last column) and mistrust rate (MTR, bottom line) for each class. These numbers reveal a slight increase in both MCR and MTR for the medium L2 level of cognitive workload indicating that a higher likelihood of misclassification for that class.

The comparison between the two confusion matrices shows no statistical difference between the two cohorts as summed up in the two test set misclassification rates of 15.17±0.79% and 17.38±1.85%. These are obtained as the ratio of off-diagonal results to the total number of experiments resulting in a slightly different outcome from the average misclassification rate due to a slight difference in the number of each experiment for each class. These numbers demonstrate that the experimental setup is just as applicable in the practical settings of pilot simulator environment as it is in the university lab.

## 5. Summary and Discussion

Assessing and monitoring the cognitive workload of pilots and air-traffic controllers is critical to managing aviation safety. Such monitoring, as previously mentioned, can be undertaken through self-assessment, task performance or measurements of physiological variables. This paper focuses on combining cardiovascular measures of pressure and heart rate with voice features so that three levels of cognitive workload can be assessed and monitored. The features used to characterise the cardiovascular system are the well defined blood pressure measures obtained from a Finometer from Finapress [43,44]. The novel voice features presented in this work are derived from the formant track features developed in previous work [16,47] and characterise the vocal tract shape and change in shape. The reason why the formant track features were chosen in contrast to of voice source features (e.g. those characterising pitch, change in pitch and other microprosodic features [50,51,52]) was that they show superior performance in cognitive workload monitoring [16] as well as displaying promise for other paralinguistic speech processing as well [47]. While promising results using more traditional speech features such as mel-frequency cepstrum coefficients have also been obtained in the literature [15] but developing these features for the data-fusion scheme proposed in this paper remains outside the scope of this study and should be considered in future work.

The classifiers used in this study are all trained on the individual participants. Consequently, a new user would have to provide training data to enroll in the system. Despite the promising results published by Yin et al. [15] with participant independent methods, individual differences have been a major confound in prior work and, hence, Yin et al.’s [15,19] results must be treated with caution.

We therefore propose a participant dependent classification scheme as the way forward. This should be supplemented with pattern recognition learning algorithms enabling the model to be further adapted to the individual. While state-of-the-art methods have achieved some progress in discriminating high and low cognitive workload, this paper moves beyond the binary paradigm by setting up a three cognitive workload level classification problem. For example, the results showed that misclassification errors were more common in adjacent classes. Many models presented in the literature [53] assume, however, that cognitive workload should be measured as a continuous variable. This presents methodological difficulties since cognitive workload is naturally a latent variable that needs to be estimated indirectly. While this paper does not solve these difficulties, these challenges are neatly demonstrated with our three class setup.

The limited vocabulary of the Stroop task, where the participants uttered the same five words throughout, might seem too limiting for real world operational scenarios. However, the features extracted are not based on linguistics but rather on the amplitude peaks in certain frequency spectra. This can therefore be the basis for a model intended for aviation personnel, limited to the vocabulary used internationally in air traffic communication terminology.

The use of two complementary psychophysiological measures are ideal. The unobtrusive speech signal gives a good measure of workload when participants are engaged in verbal communication. However, in periods with little or no speech communication, cardiovascular measurements better indicate the workload.

Although an average MCR of 15.17±0.79% may seem to be a barrier, the results in Table 2 indicate that most of the incorrect classifications lie in adjacent cognitive workload levels, e.g., high misclassified as medium. Such misclassification may indicate the boundary between adjacent levels is too rigid at a generic level and further highlight the need for granularity at the individual level. Despite these promising results, numerous challenges remain. Real time adaptation needs to be implemented and solutions for recordings of the psychophysiological signals in an unobtrusive manner devised.

Two examples of the practical implications of the methodology and results of this research relate to their application at both the strategic and tactical level, for both air traffic controllers and pilots. For the former, the importance of air traffic controller workload as the critical determinant in en-route airspace capacity has been well documented over the years, including a recent document outlining proposals for the future airspace architecture in Europe by SESAR Joint Undertaking [54]. The research methodology outlined in this paper can be used to schedule the workforce given historic workload data from individuals. By this method, the air traffic controller workforce can be scheduled in a robust manner given their workload capabilities. At a tactical level when assessing capacity, the research can be used for workload balance when required by a control sector team, in contrast to the current ad hoc, inefficient methods.

Since the 1980s, Crew Resource Management (CRM) has been developed within commercial aviation [55], with a view to reducing pilot errors and thereby aviation accident rates. CRM has its core nontechnical skills (NTS) that embrace both the interpersonal (leadership, teamwork, communication) and cognitive (decision-making, situation awareness, task management) elements necessary for safe and efficient performance [56]. CRM training has been implemented by many airlines globally and is a required part of pilot training in the US and UK [57]. For example, to be granted or maintain a valid Air Operator Certificate (AOC) in the UK, an operator must demonstrate that they can satisfy regulations mandated by the European Aviation Safety 494 Agency (EASA) and the CAA. Essential to the NTS of CRM is workload management, which is the organisation of the required flight activities to achieve goals and maintain quality and safety. This involves managing competing pressures and demands including scheduling and passenger requirements [58]. While CRM training enables pilots to manage their workload, actually understanding that workload and its causes poses challenges in this management. Hence, the methods developed in this paper can assist in this by better understanding the workload profiles of pilots and investigating the situations leading to high workload and the boundary between medium and high workload. This affords a great opportunity for the methods outlined in this paper to complement CRM to improve the management of pilot workload and therefore improve aviation safety.

## 6. Conclusions

The management of the workload of personnel in an organisation is essential for both their well being and efficiency and for the broader societal safety. The accurate measurement and monitoring of the cognitive workload is therefore a pre-requisite. To achieve any such management, however, the traditional methods of measurement based upon either self-assessed subjective ratings or on performance measures of tasks have their well-documented limitations. The use of psychophysiological measurement is another technique that has failed to gain widespread acceptance in the past due to reliability and logistical issues. However, recent advances in technology and computational abilities provide an opportunity to examine this technique anew. Whilst individual sources of psychophysiological measurement can fail to provide the reliability required for operational purposes, combining feature sets extracted from two sources may provide the requisite reliability.

In this paper, we have demonstrated the potential of using a combination the speech signal and cardiovascular measures, to measure and monitor cognitive workload. Based upon a much larger sample than in any previous study and using carefully designed experiments with increasing task complexity, we have shown that this combination can reliably measure and discriminate three levels of cognitive workload at the level of an individual. This then provides safety critical organizations with the flexibility to manage the workload of their personnel and thereby enhance safety.

## Figures and Tables

**Figure 1 sensors-22-06894-f001:**
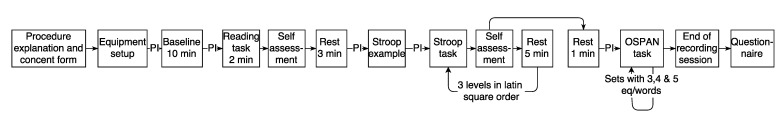
Chart of the flow of tasks and resting periods for the whole duration of one experiment. Progress Instructions PI depicts instructions given to the participants in between tasks to be read out loud.

**Figure 2 sensors-22-06894-f002:**
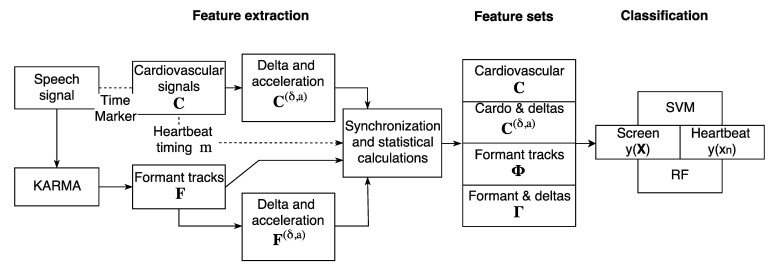
Overview of the feature extraction, feature synchronization and classification process.

**Figure 3 sensors-22-06894-f003:**
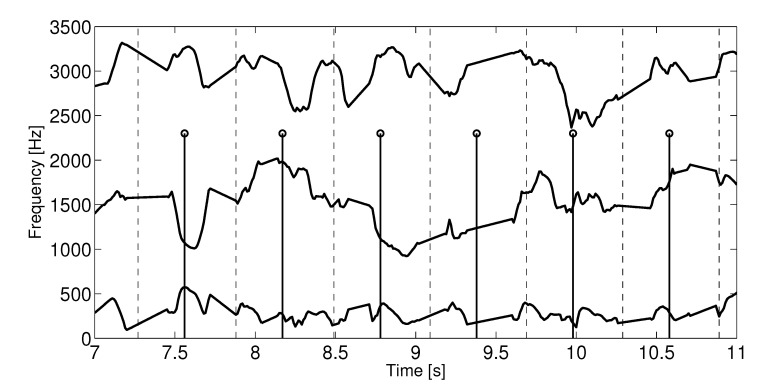
Example of the synchronization process with an approximately 10 heartbeat portion of a screen. The plot shows on a disproportional y-axis the three formant tracks. The heartbeats are indicated by the stem lines and the vertical dotted lines represent the interval on either side of the heartbeat marking the formant frames assigned to each heartbeat.

**Table 1 sensors-22-06894-t001:** Single heartbeat misclassification rate [%] for the university cohort. The best results are presented in bold.

Features	SVM	RF
$x_{n}$	$y_{SVM}\left(x_{n}\right)$	$y_{RF}\left(x_{n}\right)$
$\phi_{n}$	61.75 ± 0.18	53.50 ± 0.18
$\gamma_{n}$	57.88 ± 0.18	52.58 ± 0.18
$c_{n}$	36.62 ± 0.18	37.57 ± 0.18
$c_{n}^{\left(\delta ,a\right)}$	**36.17 ± 0.17**	37.48 ± 0.18
$\phi_{n}$ & $c_{n}$	32.57 ± 0.17	34.63 ± 0.17
$\phi_{n}$ & $c_{n}^{\left(\delta ,a\right)}$	32.43 ± 0.17	34.43 ± 0.17
$\gamma_{n}$ & $c_{n}$	32.16 ± 0.17	35.76 ± 0.17
$\gamma_{n}$ & $c_{n}^{\left(\delta ,a\right)}$	**32.14 ± 0.17**	34.57 ± 0.17

**Table 2 sensors-22-06894-t002:** Heartbeat sequence misclassification rate [%] for the university cohort. The best results are presented in bold.

Features	SVM	RF
**X**	**y** ${}_{\mathrm{SVM}}$ **(X)**	**y** ${}_{\mathrm{RF}}$ **(X)**
Φ	55.23 ± 1.10	42.07 ± 1.09
Γ	46.98 ± 1.11	41.38 ± 1.09
C	26.61 ± 0.98	**22.97 ± 0.93**
$C^{\left(\delta ,a\right)}$	25.97 ± 0.97	23.32 ± 0.94
Φ & C	17.77 ± 0.85	18.41 ± 0.86
Φ & $C^{\left(\delta ,a\right)}$	16.99 ± 0.83	19.29 ± 0.87
Γ & C	15.66 ± 0.81	19.14 ± 0.87
Γ & $C^{\left(\delta ,a\right)}$	**15.17 ± 0.79**	19.29 ± 0.87

**Table 3 sensors-22-06894-t003:** Confusion table for the sequence classification for the university cohort. Correctly classified screens are counted on the diagonal (in bold).

	Classified as			
**Actual**	**L1**	**L2**	**L3**	**MCR [%]**
**Stroop L1**	**569**	65	45	16.20
**Stroop L2**	51	**454**	77	21.99
**Stroop L3**	23	48	**705**	9.15
**MTR [%]**	11.51	19.93	14.75	**15.17**

**Table 4 sensors-22-06894-t004:** Confusion table for the sequence classification for the pilot cohort. Correctly classified screens are counted on the diagonal (in bold).

	Classified as			
**Actual**	**L1**	**L2**	**L3**	**MCR [%]**
**Stroop L1**	**113**	12	15	19.29
**Stroop L2**	6	**96**	18	20.00
**Stroop L3**	5	17	**138**	13.75
**MTR [%]**	8.87	23.20	19.30	**17.38**

## Data Availability

The data set used in this study is available at https://eyra.ru.is/coldfish/ (accessed on 29 June 2022).

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
