# Peer review of "Assessing Cognitive Workload Using Cardiovascular Measures and Voice"

_sensors, 2022, doi:10.3390/s22186894_

Round 1

Reviewer 3 Report

In this paper, a method that combines the speech signal with cardiovascular measurements for screen and heartbeat classification is introduced. For validation, speech and cardiovascular signals from 97 universities and 20 airline pilot participants were collected while cognitive stimuli of varying difficulty level were induced with the Stoop colour/word test. I think the points of this article is interesting, but still need to be refined. The specific suggestions are as follows:

1.       How is the OSPAN task mentioned in Figure 1 performed in this experiment? Please describe briefly.

2.       The title and abstract of the thesis talk about the combination of cardiovascular measurement and voice, but the conclusion is the combination of psychophysiological measures, the speech signal and cardiovascular measures. This is inconsistent.

3.       The Discussion section is too bloated, and relevant studies should not appear here.

4.  This paper only demonstrates the potential of using a combination of psychophysiological measures, speech signals and cardiovascular measures to measure and monitor cognitive workload, please add comparison experiments with state-of-the-art methods in this direction.

5.  A new feature-level fusion method between non-invasive cardiovascular measurement and speech proposed in Section 3.3, but the principle of fusion and the novelty of this fusion method are not mentioned in the specific description that follows, please add an explanation.

6.    Use OSpan sometimes (168 lines) and OSPAN sometimes (Figure 1),in the paper. Please use uniform and standard writing.

7.   In the 7th line of the Abstract, "signals from 97 university", please check the full text to avoid such grammatical errors.

Round 2

Reviewer 3 Report

I appreciate the idea of this paper and can be accepted in present form.

Author Response

Thank you for your thorough and helpful review.